# Relationship Between Internal and External Load in Under-16 Soccer Players: Heart Rate, Rating of Perceived Exertion, and GPS-Derived Variables

**DOI:** 10.3390/sports13110376

**Published:** 2025-11-03

**Authors:** Krisztián Havanecz, Sándor Sáfár, Csaba Bartha, Bence Kopper, Tamás Horváth, Péter János Tóth, Gabriella P. Szabó, Zoltán Szalánczi, Gábor Géczi

**Affiliations:** 1Training Theory and Methodology Research Center, Hungarian University of Sports Science, 1123 Budapest, Hungary; safar.sandor@tf.hu (S.S.); bartha.csaba@tf.hu (C.B.); 2Faculty of Kinesiology, Hungarian University of Sports Science, 1123 Budapest, Hungary; kopper.bence@tf.hu; 3Research Center for Sports Physiology, Hungarian University of Sports Science, 1123 Budapest, Hungary; horvath2.tamas@tf.hu; 4Department of Sport Games, Hungarian University of Sports Science, 1123 Budapest, Hungary; toth.peter.janos96@gmail.com; 5Department of Performance Diagnostics Research, National Institute for Sports Medicine, 1113 Budapest, Hungary; dr.p.szabo.gabriella@osei.hu; 6Faculty of Economics and Business, University of Debrecen, 4032 Debrecen, Hungary; szalanczi.zoltan@econ.unideb.hu; 7Department of Sport Management, Hungarian University of Sports Science, 1123 Budapest, Hungary; geczi.gabor@tf.hu

**Keywords:** GPS, heart rate, monitoring, RPE, training load, youth soccer

## Abstract

Heart rate (HR) monitoring is a practical method for assessing internal load (IL). However, it remains unclear for which age group HR would be an appropriate predictor of IL considering the relationship with external load (EL). Thus, this study aims to evaluate the relevance and applicability of HR monitoring by exploring the relationship between EL and IL among U16 soccer players. EL was measured using global positioning system (GPS) data, while IL was assessed through training impulse (TRIMP), Edward’s TRIMP, HR exertion, rate of perceived exertion (RPE), and session-RPE (s-RPE). Nineteen (N = 19) male footballers from an elite football academy participated, with data collected from 50 training sessions and 11 matches. In the analysis of the training sessions, TRIMP demonstrated a near-perfect correlation with total distance (TD) (*p* < 0.001), and eTRIMP correlated strongly with TD (r = 0.82) and player load (r = 0.79). HR exertion also correlated significantly with TD, medium-speed running, decelerations, inertial movement analysis (IMA) events, and player load (*p* < 0.001). In matches, a large correlation was observed between TRIMP and TD (r = 0.73), while the strongest correlation was between RPE and s-RPE with TD and PL (*p* < 0.001). Furthermore, TD emerged as the best GPS-derived predictor of both TRIMP and HR exertion in training contexts. These findings provide evidence for the validity and usability of heart rate-based and RPE-based measures to indicate IL in U16 soccer players. Future research should focus on contextual factors in exploring the relationship between EL and IL.

## 1. Introduction

Training load (TL) has been defined as the physiological and psychological stress experienced by an individual [1] and is used to elicit the desired training response by manipulating training variables [2]. TL can be further divided into external load (EL) and internal load (IL) [3]. In soccer, EL refers to the physical movements of a player, which can be expressed in various running speed zones, accelerations, and decelerations [4,5], while IL encompasses the psycho-physiological stress imposed on a player [6].

The global positioning system (GPS) is being recognized as a standard device for quantifying and monitoring EL in soccer [7] and provides reliable locomotor TL variables. The use of integrated microtechnology, commonly known as inertial measurement units (IMUs), provides mechanical TL variables derived from a tri-axial accelerometer, magnetometer, and gyroscope [8]. The current challenge is to identify the relevant GPS-based variables in terms of TL regarding a specific sport. Based on scientific evidence and citations, it has been suggested to use total distance, high-speed running, sprint running, accelerations, decelerations, and player load in soccer [9,10,11,12]. Nowadays, researchers use both volume and intensity parameters to precisely determine players’ EL [13,14].

Conversely, heart rate (HR) serves as an individually based yet objectively measured [15] indicator of IL, offering valid and reliable feedback on players’ physiological responses [16] during both training sessions and matches [17]. However, compared with adults for young soccer players, the natural growth and maturation of the heart is an ongoing process, making it challenging to accurately determine their internal TL [18]. To record HR, a wearable technology is required [19]; therefore, in soccer, for monitoring this a non-invasive method is universally employed [20]. Interestingly, commercially available photoplethysmography-based devices (sports watches) are not allowed on the field; instead, ECG-based heart rate sensors have become prevalent in soccer. Previous studies have examined heart rate data using TRIMP (total duration multiplied by average heart rate), Banister’s TRIMP [21], which assumes that training load is equal to exercise duration multiplied by exercise intensity [22], and Edward’s TRIMP [23], which is calculated by multiplying the time spent in five HR zones by arbitrary coefficients [22]. These equations are used for field-based training and also for matches [22,24]. However, Banister’s TRIMP is limited by the difficulty of accurately measuring resting HR [25], as players arrive on the field in an elevated physiological and psychological state. Furthermore, it was found that HR measurements could be inadequate to determine IL during very-high intensity training [26]. Instead, it is recommended to use Edwards’ TRIMP, which provides a more accurate representation of TL [19,22,27] by considering exercise intensity.

Additionally, the rating of perceived exertion (RPE), a subjective measure of exercise intensity, and session-RPE (s-RPE), calculated by multiplying the RPE score by the session duration, have been widely employed in soccer to quantify the individual perception of IL [14,28,29,30]. Previous research has reported moderate to very large correlations between Edward’s TRIMP and s-RPE, suggesting their validity and reliability [22,31].

In elite soccer, GPS and HR monitoring technologies are commonly used together to assess EL and IL [32], representing the cumulative exposure of each player to training and competition [33]. However, this topic has not been investigated in youth soccer, and the appropriate starting age for monitoring has yet to be defined. It is challenging to ensure reliable measurement of HR in youth soccer players, as the recorded values may reflect sympathetic nervous system responses influenced by external environmental factors rather than providing an accurate representation of TL. Therefore, the primary aim of this article is to explore the feasibility of tracking heart rate for a U16 age group in soccer by examining the relationship between HR-based metrics, GPS-derived variables with RPE, and s-RPE data. We also sought to identify GPS variables that could predict HR values in trainings.

## 2. Materials and Methods

### 2.1. Procedure

A total of 50 field-based training sessions and 11 matches were conducted during the study, resulting in over 534 observations (465 training, 69 matches). We have intentionally focused on the competition period for our data collection, as training adaptation could otherwise influence the results. We only considered training session data for weeks with at least one match. The training sessions were performed on weekdays at the same time of the day, roughly between 15:30 and 17:00, at least four times a week on artificial pitches. During both training sessions and matches, GPS-derived and HR-based data were collected. Data from warm-up sessions, which lasted approximately ten minutes before ball training, were also included in the analysis. Additionally, training sessions that occurred indoors were excluded due to satellite connection errors, which may affect the reliability of the GPS-based variables.

The training program consisted of both technical and tactical components. The warm-up followed the FIFA 11+ protocol [34] and was included in the data analysis alongside ball training, while the cool-down period was excluded. The microcycle structure was defined as a weekly cycle within an annual program [35]. Training days were referenced relative to match day (MD±), in accordance with previous literature [36]. All teams had rest days on MD-6 and MD+1. MD-5 and MD+2 were designated as recovery sessions that also included tactical development, which included a longer warm-up with and without the ball, with overload small-sided games (SSGs), such as 4v2 or 5v2 two-zone possession formats. On MD-4 and MD+3, the training focus was on SSGs (e.g., 1v1, 2v2, 3v3) to enhance speed and reactive agility. These drills were characterized by reduced overall duration but involved high-intensity intermittent bouts of efforts with sufficient rest times. MD-3 targeted strength-endurance through large-sided games with and without specific constraints (e.g., free play), which was executed with small overload (e.g., 8v7, 9v7) or with balanced participants (e.g., 9v9, 10v10). MD-2 emphasized technical and tactical preparations specifically based on the upcoming opponent and was the training day with the lowest TL in terms of intensity. MD-1 served as an “activation” day with neuromuscular focus and reduced training load in terms of duration. These trainings involved more reaction-based SSGs, and on regular basis, it was the shortest session of the week. Additionally, a 30 min strength and conditioning session was implemented on Mondays, Tuesdays, and Thursdays prior to ball training.

### 2.2. Participants

Nineteen (N = 19) male academy soccer players (age 15.4 ± 0.4 years, age at peak height velocity 14.4 ± 0.4 years, maturity offset 1.4 ± 0.5 years, height 175.3 ± 7.2 cm, body mass 61.7 ± 5.2 kg; all measurements are mean ± standard deviation) participated in this study. Inclusion criteria required that only outfield players who attended at least 80% of training sessions and completed at least 60 min in a match were included [37]. Another necessary criterion was that only participants with no known injuries during the study were included. Goalkeepers were not included in the present study due to the lack of sensors to wear. Since confounding factors such as dehydration may influence players’ perceived exertion, players were allowed to drink still water during training sessions’ resting periods. Considering matches, fluid intake was uncontrollable for the players on the field, and they had time to replenish at the half time with certainty. All participants were informed about the goal of the study and were already familiarized with the appropriate devices to wear (at the start of pre-season) as part of their daily routine monitoring, and the written consent of their parents or legal guardians was obtained. This research received institutional ethical approval and was conducted in accordance with the latest version of the Declaration of Helsinki.

### 2.3. External Training Load Measurement Procedures

EL data was collected using a 10 Hz portable GPS device (Catapult Vector S7, Catapult Sports Ltd., Melbourne, Australia) equipped with an integrated 100 Hz tri-axial accelerometer, magnetometer, and gyroscope. The combination of these components has demonstrated acceptable validity and reliability for measuring distance and high-speed movement metrics in team sports contexts [38], with a typical measurement error reported at 1.3% [39]. To minimize inter-unit variability, all players wore the same model of GPS device throughout the study [40]. The average horizontal dilution of precision (HDOP) during data collection was 0.76 ± 0.1, with an average of 14.88 ± 0.2 connected satellites and a mean Global Navigation Satellite System (GNSS) signal quality of 70.51% ± 1.1%; these values are accordance with previously published standards [41]. Data sets showing substantial deviation from these signal quality indicators were excluded from the analysis. Each device was worn in a manufacturer-provided vest (Catapult Sports) positioned between the scapulae. In line with the recommendations of Maddison and Ni Mhurchu (2009) [42], all units were activated while docked at least five minutes prior to the start of each session to ensure stable satellite connection and optimal signal acquisition. The selected locomotor and mechanical TL variables are presented in Table 1. The velocity-based variables were classified based on previous literature [43,44], which later become the default setting in the Catapult system.

The player load variable was determined using the following equation expressed in AU [45]:$$PL=\sqrt{\frac{{(a}_{y1}-a_{y-1}{)}^{2}+{(a}_{x1}-a_{x-1}{)}^{2}+{(a}_{z1}-a_{z-1}{)}^{2}}{100}}$$
where, $a_{y}$ denotes forward/backward accelerometer, $a_{x}$ denotes sideways accelerometer, and $a_{z}$ denotes vertical accelerometer [45].

### 2.4. Internal Training Load Measures

Heart rate was recorded using a Polar heart rate sensor (Polar H9, Polar Electro Oy, Kempele, Finland). The device was attached to a chest strap positioned between the abdomen and the chest, a little bit above the processus xiphoideus. It was worn together with the Catapult vest and activated simultaneously and synchronized with the GPS sensor via Bluetooth. At the beginning of the pre-season, the individual maximum heart rate (HR_max_) was determined through a maximal incremental spiroergometric running test, which could later be modified during training and/or matches (new records can be updated in the OpenField Cloud system after each activity). After data collection, IL was expressed using three formulas (see Table 2). For the HR exertion measure, eight different HR zones were established based on the default settings of the Catapult system. The method of Edwards’ TRIMP (eTRIMP) was employed [23]. This expresses the heart rate (HR) responses of athletes as a percentage of their HR_max_. In this context, eTRIMP was calculated based on the time spent in the five HR zones, multiplied by a zone-specific weighting factor.

For the purposes of this study, we also included the rating of perceived exertion (RPE) to assess players’ subjective exertion, as proposed by Foster (1998) [46]. It represents a single number expressed in arbitrary unit (AU) to calculate TL. It was asked 20–30 min after the activities [47] by the team’s strength and conditioning coach in person and later recorded in an Excel spreadsheet. A modified Borg CR-10 (Category-Ratio) scale was used, where players were informed that ‘1’ indicates very weak activity and ‘10’ indicates extraordinarily strong activity [48]. Also, to derive session-RPE (s-RPE), the RPE was multiplied by total duration in minutes [47] after each training session and match. If a player failed to respond in time (<1 h after an activity), even if he later did provide information, that data was discarded from further analysis due to bias of subjective feeling of fatigue.

### 2.5. Statistical Analysis

All results are presented as means and standard deviations (mean ± SD). Training sessions and matches were analyzed separately. Following each physical session, GPS- and HR-based data were collected using the OpenField Console software (Catapult Sports, Melbourne, Australia; version 3.9) and subsequently exported in .csv format for further analysis. Statistical analyses were performed using JASP software (The Jeffrey’s Amazing Statistics Program; version 0.19.2, JASP Team, Amsterdam, The Netherlands). Descriptive statistics were calculated, and the Shapiro–Wilk test was employed to assess the normality of the data. As the data were not normally distributed, Spearman’s rank correlation coefficient was used to examine associations between EL and IL metrics. The strength of correlations was interpreted according to the thresholds proposed by Hopkins et al. (2009) [49]: trivial (r < 0.1), small (0.1 ≤ r < 0.3), moderate (0.3 ≤ r < 0.5), large (0.5 ≤ r < 0.7), very large (0.7 ≤ r < 0.9), and almost perfect (r ≥ 0.9). In addition, HR-derived variables from trainings were entered into a multiple linear regression model to investigate predictors of IL. The GPS-based variables included as predictors—TD, HSR, SPR, and ACC—were selected. Model performance was evaluated using standardized beta coefficients, R-squared values, adjusted R-squared, and root mean of standard error (RMSE). We considered variance inflation factor (VIF) values between 5 to 10 and a tolerance value of less than 0.20 as indicative of potential multicollinearity problems. Statistical significance was set at *p* < 0.05.

## 3. Results

Table 3 shows the descriptive statistics of EL and IL variables dataset.

Figure 1 illustrates weekly variation in IL and EL metrics during training sessions and matches over an 11-week period. After the evaluation of the datasets, TRIMP and HR exertion demonstrated similar pattern, indicating a strong correlation between these two variables of IL, particularly when analyzed as weekly averages throughout the training period. Similarly, TD and PL exhibited a comparable trend, with the exception of deviations observed in weeks two and eleven. In contrast, the TL variables observed during matches were less pronounced, as weeks two, three, eight, nine and eleven demonstrated a reduced strength of association between the variables.

Figure 2 presents the relationship between IL regarding both subjective and objective parameters. For both trainings and matches, the strongest relationship was observed between TRIMP and HR exertion. Additionally, s-RPE demonstrated a very large correlation with TRIMP (*p* < 0.001).

Figure 3 and Figure 4 present the correlations between EL and IL volume and intensity parameters during training sessions and matches. Considering trainings, HR exertion showed a large correlation with TD, MSR, DEC, IMA, and PL (*p* < 0.001). Its intensity variable (HR exertion/min) exhibited a small to moderate correlation with GPS-based variables. Notably, TRIMP showed a near-perfect correlation with TD (*p* < 0.001) and very large correlation with MSR, DEC, IMA, and PL ranging from 0.72 to 0.88. Additionally, eTRIMP exhibited very large correlation with TD and PL. Interestingly, RPE and s-RPE showed very large correlation with TD and PL (*p* < 0.001).

In matches, TRIMP demonstrated a very large correlation with TD from HR-based variables (*p* < 0.001). Furthermore, RPE and s-RPE showed a very large correlation with TD.

Meanwhile, multiple linear regression analysis was conducted to evaluate the extent to which GPS variables could predict HR metrics. Low collinearity was confirmed with tolerance values ranging from 0.38 to 0.78 and VIF ranging from 1.28 to 2.73. A significant regression was found (*F*(1, 465) = 596.71, *p* < 0.001) with an R^2^ of 0.82. The average TRIMP score for U16 players increased for each meter of TD (ß = 0.98, *p* < 0.001), while other covariates such as HSR, SPR, and ACC contributed minimally to the model showing small to very small positive effect. A significant correlation was found (*F*(1, 465) = 1156.49, *p* < 0.001), indicating that TD explained 71% of the variance in HR exertion. Additionally, standardized beta coefficients indicated that HSR, SPR, and ACC were the weakest predictors with negligible effect on HR exertion. Considering RMSE, the results indicate an acceptable level of predictive accuracy with an error of approximately 12% of the mean TRIMP when predicted from total distance, and around 17% of the mean HR exertion when predicted from total distance.

## 4. Discussion

The primary aim of this study was to identify the relationship between HR metrics and GPS-derived variables for U16 youth male soccer players. Monitoring TL can play a crucial role in optimizing the physical preparation of youth soccer players and provide valuable support for coaches in individualizing training stimuli. A recent study was conducted with the U16 age group, analyzing half a year of training and match data separately, and differences in weekly mean TLs were assessed using descriptive statistical analysis.

Upon examining the descriptive statistics, variations were observed in both EL and IL across different weeks, in both trainings and matches. In terms of correlation results, we found the strongest relationship between EL and IL concerning volume variables of TD and PL with TRIMP. Recent studies on soccer have reported large associations between TRIMP or Edwards-based TRIMP and TD and PL, supporting the role of volume-based HR metrics as valid indicators of overall cardiovascular load [50]. However, notable deviations were detected in weeks 2 and 11, where athletes appeared to engage in more high-intensity activities. Although this was not directly analyzed, the IL values suggest an increased intensity during these weeks, further highlighting the limitations of HR-based measures in accurately monitoring high-intensity efforts [26]. Experimental and field studies have reported poor agreement between HR and RPE or sport-specific drill intensities during high-intensity intermittent activities. Our findings therefore support the notion that TRIMP may be less sensitive to transient, anaerobic-intensive spikes in TL [51]. On the other hand, eTRIMP showed strong correlation with TD and PL, which is consistent with the findings of previous literature [19,22]. Our results also support the conclusions of Lisbôa et al. [26] that higher intensity training can affect HR; thus recommending the use of eTRIMP to manage this sensitivity [19,22]. Thus, it may offer advantages for youth monitoring, especially when sessions include frequent high-intensity bouts [22]. Moreover, HR exertion showed an extraordinarily strong correlation with TD, MSR, DEC, IMA, and PL variables. Unfortunately, there is limited research literature regarding this subject, but the reliability of the HR exertion variable is acknowledged in the context of matches [52]. Regarding intensity variables, TD/min showed a large correlation with all heart rate-based variables (*p* < 0.001). This confirms the importance of examining intensity parameters in TL research [13]. The weakest relationship was observed with ACC/min and IMA/min, suggesting that these variables should be further examined. In this study, each intensity zone was integrated into a formula, yielding weak results; therefore, it is advisable to apply them separately [30].

In matches, we found a very strong correlation between TRIMP and TD, reinforcing the significance of these two variables in determining TL [27]. The most pronounced differences were observed in weeks 3 and 11. Matches in these weeks were substantially less physically demanding compared to others, as indicated by lower IL values despite relatively high EL, suggesting a mismatch between physical output and perceived exertion. It has been previously proposed that the intensity of matches is superior to that of training [2], which is in concordance with our findings regarding HR values, making them much more sensitive [22]. These considerations may account for the observed small to moderate correlations between eTRIMP and HR exertion/min, and EL metrics such as TD and PL. Furthermore, s-RPE emerged as a more reliable indicator of IL intensity. Prior literature has similarly reported that match demands are often qualitatively different than training (e.g., higher peak-intensity actions), which complicates simple one-to-one mapping between HR- and GPS-derived metrics [22]. Therefore, TL management is not the last consideration in training planning for youth soccer players.

The individual characteristics indicate that the players who were at the post-PHV period were close to biological maturation. The strong coupling observed between TRIMP/HR exertion and TD is therefore consistent with players whose growth trajectories have largely stabilized. Previous studies suggest that maturation influences the relationship between EL and IL. Previous study highlighted that maturation influences how players respond to load (e.g., shifts in HR response) and may cause moderate associations [53]. Another similar study found that players with more advanced maturity status (close to adulthood) achieved higher performance in running metrics and demonstrated different IL responses, pointing to the influence of maturation on load tolerance and physiological efficiency [54]. Comparatively, in older youth and adult players (e.g., Under 17 to Under 19 population) studies have further shown consistent EL and IL relationships in matches, reinforcing that as biological maturation stabilizes, the link between EL and IL becomes more robust. Thus, contextualizing our results, it suggests that our observed strong EL and IL correlations are plausible for post-PHV U16 players and are broadly consistent with findings in older adolescent soccer players. Consequently, it is worthwhile to monitor these variables starting from this age group, along with HR-based metrics.

The secondary aim was to identify the best IL predictor via GPS-based variables during practice. Based on our linear regression model, we found that TD is the best predictor of HR exertion and TRIMP variables (*p* < 0.001). This result strongly corroborates with previous research [19] and further supports the close relationship between TD and IL parameters for youth soccer [13,14,32]. That said, models that include intensity-specific variables (e.g., high-speed running, sprint distance, accelerations) can improve prediction of perceived exertion or match-specific strain in many contexts, especially when sessions contain high-velocity or repeated-sprint bouts [55].

Additionally, we correlated s-RPE with HR-based variables (TRIMP, HR exertion) and found a very large relationship, further validating the s-RPE method in youth soccer considering both trainings and matches [14,30,56]. In youth cohorts specifically, s-RPE has repeatedly been shown to correspond well to GPS-derived volume metrics, reinforcing its utility as a complementary tool when HR monitoring is impractical or when perceived/psychological load is needed [56].

### 4.1. Practical Applications

The findings of the research offer significant contributions to the understanding of the relationship between EL and IL in U16 soccer. In a training context, HR-based variables appear to show a stronger association with GPS-derived metrics compared with matches. When assessing IL through HR measures, it is recommended to consider multiple, composite variables rather than focusing solely on simpler metrics such as the current beat-to-beat value (HR) and the average rate of HR. For tracking TL in real time, cumulative indices such TRIMP, eTRIMP, and HR exertion may offer a more accurate representation of IL and thus give the advantage to better control the TL regarding training programs. For instance, if a player presents with a cumulative TL exceeding the recommended threshold by MD-2, it is advisable to implement differentiated training programs, such as the tapering method, in order to optimize readiness for the upcoming match. Conversely, if a player has accumulated an insufficient TL by MD-2, additional conditioning (e.g., additional running drills or compensatory exercise blocks) may be prescribed to ensure adequate preparation and reduce the risk of undertraining and overall, the occurrence of non-contact injury for matches. However, during matches, significantly higher correlations were observed with RPE and s-RPE, which may indicate its utility. Given the limited or non-availability of HR monitoring tools in many football academies, there is a practical need for an accessible, cost-effective and easy-to-interpret measurement methodology, which the RPE scale effectively fulfills. While subjective measures cannot replace standardized HR metrics, in certain circumstances, it is possible to implement alternative solutions as proxies. In practice, we recommend that data collection be conducted by a practitioner, following established recommendations in the literature. Delayed self-reporting by players is likely to reduce the reliability of the data, as responses obtained in a resting state may underestimate the actual exertion experienced shortly after physical training. For future research directions, it is recommended to prioritize and emphasize the most relevant TL variables, as a greater volume of data does not necessarily translate into improved insights or outcomes.

### 4.2. Limitations

While the study provides forward-looking insights into the relationship between GPS-derived variables, HR metrics, and RPE in youth soccer, there are, however, several limitations that should be mentioned. Firstly, we are aware of the fact that the simple size is limited, as only the team of U16 players participated in the current study from one football academy. However, this was an intentional methodological consideration to ensure sample homogeneity. Although players were not exposed to the same TL considering both EL and IL, they participated in the same training sessions, thereby reducing the possibility of deviation in the influence of independent external factors such as environment, play style, rules, and coach attitude on the individuals in the sample.

Secondly, the shortcomings in the contextual factors (e.g., environmental conditions, relative speed zones, opponent in matches, overtraining) that may influence the relationship between EL and IL. At the beginning of the season, mean ambient temperature was high, whereas by the middle of the season it had consistently declined, representing a substantial environmental change, which may influence players’ perceptual responses to RPE. Also, among the U16 players, there may have been some who were in late puberty, so in their case, executing a sprint could be challenging.

Thirdly, a breakdown by match days would show additional interesting results, but we did not consider this due to the small dataset. However, different training days have different TL values, so the relationship between EL and IL may vary.

Also, TD emerged as the main predictor of IL, and this finding is consistent with previous research. However, the analyses did not account for the repeated-measures structure of the data, and speed and acceleration thresholds were not individualized, which may have affected the precision of EL estimation.

Lastly, it is important to emphasize that trimming GPS data may significantly influence intensity-related parameters, as it matters where each period within an activity begins and ends. In this study, all periods were analyzed in activities (training sessions), meaning that rest intervals were also included. These often involve tactical discussions, carrying football goals, changing jerseys, and refreshing with water. In the context of matches, such interruptions typically occur in the form of stoppages (e.g., standing kicks, substitutions, cooling breaks due extreme heat, or sudden injuries), although these are generally less frequent in the latter cases.

## 5. Conclusions

This study provides valuable insights into the relationship between EL, as measured by GPS-derived variables, and IL, represented by HR-based metrics, RPE, and s-RPE, in U16 soccer. The findings revealed large to almost perfect correlations between EL and IL measures, with TD and PL emerging as the most strongly associated GPS metrics with TRIMP and HR exertion during training. Notably, TD proved to be the most consistent predictor of both TRIMP and HR exertion, underscoring its relevance in monitoring TL. These results are particularly meaningful for practitioners, as TD is commonly used by coaches to evaluate training demands. Furthermore, strong correlations were observed between TD, PL, and RPE and s-RPE, suggesting perceived exertion serves as a reliable substitute for IL assessment, especially in settings where HR monitoring is not feasible. These results are reinforcing the practical utility of GPS technology, HR monitoring, and RPE measures in U16 soccer for effective TL management and the physical development of the players.

## Figures and Tables

**Figure 1 sports-13-00376-f001:**
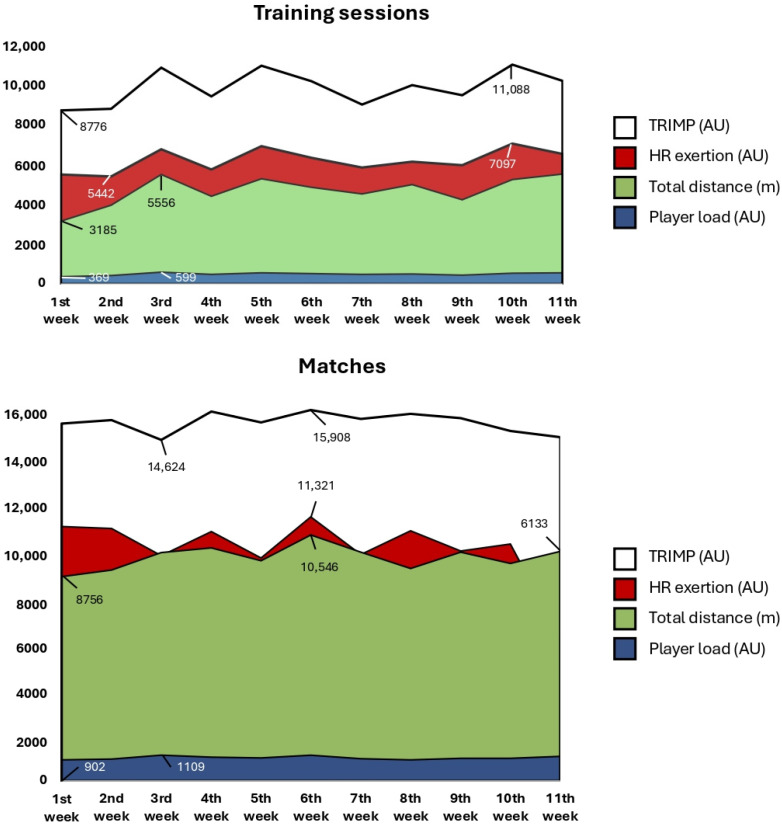
Weekly analysis of average HR-based IL parameters and GPS-derived EL metrics regarding trainings and matches. Minimum and maximum values are presented. AU = arbitrary unit, m = meter, TRIMP = training impulse, HR exertion = heart rate exertion, TD = total distance, PL = player load.

**Figure 2 sports-13-00376-f002:**
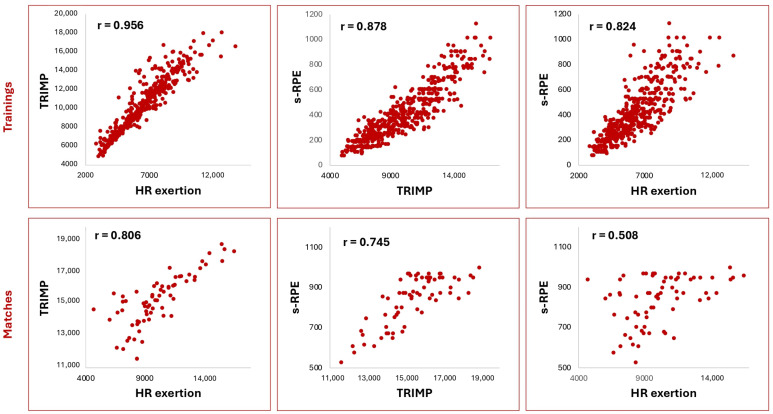
The relationship between HR exertion, TRIMP, and s-RPE in training and matches. TRIMP = training impulse, HR exertion = heart rate exertion, s-RPE = session-rating of perceived exertion.

**Figure 3 sports-13-00376-f003:**
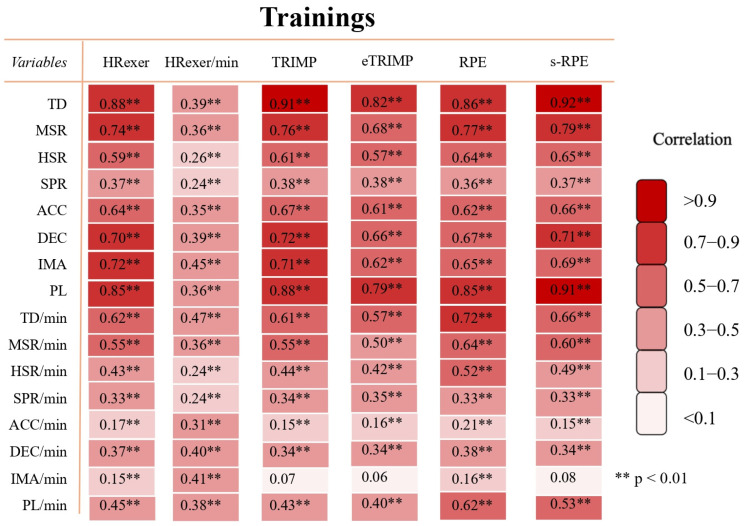
Relationship between GPS-derived variables, HR-based metrics, RPE and s-RPE measures regarding training sessions. Values represent correlation coefficients between corresponding variables. Color code indicates strength of correlation between variables: trivial (r < 0.1), small (0.1 ≤ r < 0.3), moderate (0.3 ≤ r < 0.5), large (0.5 ≤ r < 0.7), very large (0.7 ≤ r < 0.9), and almost perfect (r ≥ 0.9). Confidence interval of 95% was between 0.35 and 0.95 with significant correlations. HREXER = heart rate exertion, TRIMP = training impulse, eTRIMP = Edward’s TRIMP, RPE = rating of perceived exertion, s-RPE = session-RPE, TD = total distance, MSR = medium-speed running distance, HSR = high-speed running distance, SPR = sprint distance, ACC = accelerations, DEC = decelerations, IMA = inertial movement analysis, PL = player load. Legend: ** indicates *p* < 0.01.

**Figure 4 sports-13-00376-f004:**
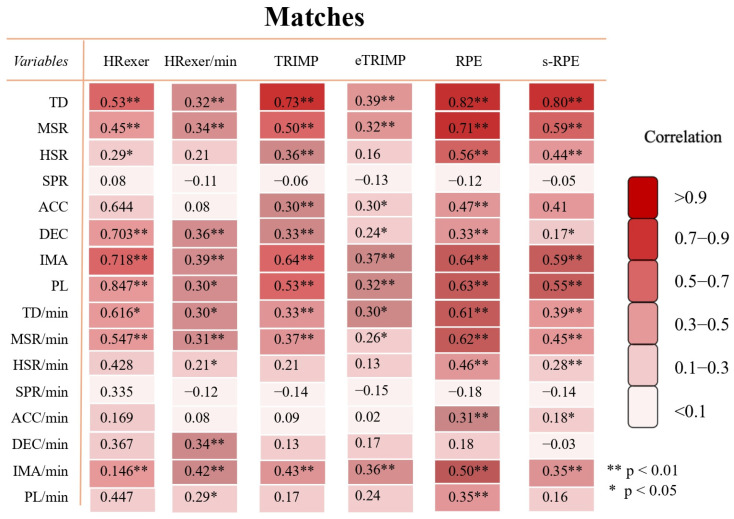
Relationship between GPS-derived variables, HR-based metrics, RPE and s-RPE measures regarding matches. Values represent correlation coefficients between corresponding variables. Color code indicates strength of correlation between variables: trivial (r < 0.1), small (0.1 ≤ r < 0.3), moderate (0.3 ≤ r < 0.5), large (0.5 ≤ r < 0.7), very large (0.7 ≤ r < 0.9), and almost perfect (r ≥ 0.9). Confidence interval of 95% was between 0.31 and 0.65 with significant correlations. HREXER = heart rate exertion, TRIMP = training impulse, eTRIMP = Edward’s TRIMP, RPE = rating of perceived exertion, s-RPE = session-RPE, TD = total distance, MSR = medium-speed running distance, HSR = high-speed running distance, SPR = sprint distance, ACC = accelerations, DEC = decelerations, IMA = inertial movement analysis, PL = player load. Legends: ** indicates *p* < 0.01, * indicates *p* < 0.05.

**Table 1 sports-13-00376-t001:** GPS-derived TL variables.

	Locomotor TL Variables	Mechanical TL Variables
Volume parameters	TD (m)	ACC (m)
	MSR (m)	DEC (m)
	HSR (m)	IMA (n)
	SPR (m)	PL (AU)
Intensity parameters	TD/min (m)	ACC/min (m)
	MSR/min (m)	DEC/min (m)
	HSR/min (m)	IMA/min (n)
	SPR/min (m)	PL/min (AU)

TL = training load, m = meter, n = number, AU = arbitrary unit, TD = total distance, MSR = medium-speed running distance (14.4–19.8 km·h^−1^), HSR = high-speed running distance (19.8–25.2 km·h^−1^), SPR = sprint distance (>25.2 km·h^−1^), ACC = accelerations (>1.5 m·s^−2^), DEC = decelerations (<−1.5 m·s^−2^), IMA = inertial movement analysis, PL = player load.

**Table 2 sports-13-00376-t002:** Formulas showing the players’ internal training load.

Methods	Equations
TRIMP	average heart rate × total duration expressed in AU
eTRIMP	time spent in zone 1 (50–59% of HR_max_) multiplied by 1, time spent in zone 2 (60–69% of HR_max_) multiplied by 2, time spent in zone 3 (70–79% of HR_max_) multiplied by 3, time spent in zone 4 (80–89% of HR_max_) multiplied by 4, time spent in zone 5 (90–100% of HR_max_) multiplied by 5, and these scores were subsequently summed and expressed in AU
HR exertion	time spent in zone 1 (≤45% of HR_max_) multiplied by 1, time spent in zone 2 (45–55% of HR_max_) multiplied by 1.122, time spent in zone 3 (55–65% of HR_max_) multiplied by 1.322, time spent in zone 4 (65–75% of HR_max_) multiplied by 1.554, time spent in zone 5 (75–85% of HR_max_) multiplied by 2.037, time spent in zone 6 (85–95% of HR_max_) multiplied by 3.252, time spent in zone 7 (95–105% of HR_max_) multiplied by 5.439, time spent in zone 8 (>105% of HR_max_) multiplied by 9.0, and these scores were summarized and expressed in AU

HR = heart rate, HR_max_ = maximal heart rate, TRIMP = training impulse, eTRIMP = Edward’s training impulse, AU = arbitrary unit.

**Table 3 sports-13-00376-t003:** Descriptive statistics of EL and IL variables in trainings and matches.

	Trainings		Matches	
	Mean ± SD	Min–Max	Mean ± SD	Min–Max
External TL variables				
TDu	1:10:49 ± 0:17:28	0:39:46–1:53:10	1:32:03 ± 0:06:24	1:15:37–1:40:19
TD	4893.2 ± 1904.1	1461.7–10,020.1	9582.2 ± 1214.3	6571.8–13,187.8
MSR	428.4 ± 309.2	39.2–1559.5	1382.9 ± 448.6	683.1–2656.5
HSR	115.4 ± 109.3	0.0–628.1	474.3 ± 160.1	175.1–797.0
SPR	17.4 ± 35.7	0.0–210.0	85.1 ± 53.8	0.0–249.2
ACC	188.5 ± 79.7	32.4–479.2	313.5 ± 68.8	166.8–444.9
DEC	70.6 ± 33.8	12.1–188.8	128.0 ± 34.2	50.7–217.0
IMA	398.5 ± 134.9	115.0–890.0	489.5 ± 159.9	279.0–972.0
PL	519.3 ± 193.9	168.8–1126.3	992.7 ± 182.2	686.4–1633.6
TD/min	67.3 ± 13.7	36.8–100.5	104.8 ± 10.0	87.1–138.5
MSR/min	5.7 ± 3.2	0.7–17.0	15.1 ± 4.6	7.9–27.9
HSR/min	1.5 ± 1.2	0.0–7.2	5.2 ± 1.7	2.5–8.9
SPR/min	0.2 ± 0.5	0.0–4.5	0.9 ± 0.6	0.0–2.7
ACC/min	2.6 ± 0.9	0.5–6.2	3.4 ± 0.7	1.9–4.8
DEC/min	1.0 ± 0.4	0.2–2.5	1.4 ± 0.4	0.8–2.3
IMA/min	5.6 ± 1.5	2.1–10.3	5.3 ± 1.6	3.5–10.3
PL/min	7.2 ± 1.5	3.6–12.2	10.9 ± 1.8	7.4–17.1
Internal TL variables				
TRIMP	10,138.6 ± 2837.2	4970.1–18,104.9	15,402.9 ± 1616.3	11,590.9–18,921.2
eTRIMP	160.2 ± 33.4	91.0–227.0	222.5 ± 67.5	92.0–396.0
HR exer	6341.5 ± 1968.5	2813.9–13,786.9	10,131.3 ± 2528.9	4667.9–16,575.7
HR exer/min	88.7 ± 14.1	54.9–144.5	110.8 ± 25.0	49.9–172.7
RPE	4.6 ± 1.7	1.0–9.0	9.1 ± 0.8	8.0–10.0
s-RPE	423.7 ± 217.1	80.0–1130.0	836.8 ± 121.3	528.0–1000.0

TL = training load, sd = standard deviation, min = minimum, max = maximum, TDu = total duration (hours:minutes:seconds), TD = total distance (meter), MSR = medium-speed running distance (meter), HSR = high-speed running distance (meter), SPR = sprint distance (meter), ACC = accelerations in distance (meter), DEC = decelerations in distance (meter), IMA = inertial movement analysis (score), PL = player load (arbitrary unit), TRIMP = training impulse (arbitrary unit), eTRIMP = Edward’s TRIMP (arbitrary unit), HR exer = heart rate exertion (arbitrary unit), RPE = rating of perceived exertion (arbitrary unit), s-RPE = session-RPE (arbitrary unit).

## Data Availability

The original contributions presented in the study are included in the article, further inquiries can be directed to the corresponding author/s.

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
