# Peer review of "Relationship Between Internal and External Load in Under-16 Soccer Players: Heart Rate, Rating of Perceived Exertion, and GPS-Derived Variables"

_sports, 2025, doi:10.3390/sports13110376_

Round 1
Reviewer 1 Report
Comments and Suggestions for Authors
Dear Editor,
Thank you for the opportunity to review the manuscript entitled “Relationship between Internal and External Load in U-16 Soccer Players: Heart Rate, Perceived Exertion Assessment, and GPS-Derived Variables.” The study followed a U-16 academy squad over an 11-week period (50 training sessions and 11 matches), combining internal load indicators (heart rate metrics and s-RPE) with external load obtained from GPS monitoring.
The research question is relevant within the context of youth soccer. However, the originality of the contribution should be addressed with more caution. Similar approaches linking GPS-derived metrics, heart rate, and perceived exertion have already been reported, particularly in elite youth soccer. In this case, the novelty seems to rely mainly on the age category and the use of the HR-effort variable, which by itself may not substantiate a strong claim of innovation.
The introduction provides some context but does not present explicit, testable hypotheses. A clearer definition of expectations, grounded in evidence specific to this age group, would reinforce the rationale for the study. From a methodological standpoint, the monitoring devices and variables are appropriate, yet some important operational details are missing. For example, thresholds for high-speed running, sprinting, accelerations, and decelerations are not reported. Likewise, the HR-effort variable is introduced only through the manufacturer’s terminology, without a technical explanation or citation of validation studies, which limits both replication and comparison with existing literature.
The statistical approach raises further concerns. Considering the repeated measures within players, simple correlations and linear regressions are not the most suitable methods. Mixed-effects models or repeated-measures correlations would provide more robust insights. Additionally, the regression analyses lack transparency regarding variable selection, and the focus on total distance tends to overshadow other potentially meaningful variables. Comprehensive model outputs, including coefficients, confidence intervals, and diagnostic checks, would be necessary to strengthen the findings.
The presentation of results would also benefit from greater clarity and structure. Confidence intervals are seldom reported, and important details related to data processing, such as GPS quality criteria, treatment of warm-up data, or whether goalkeepers were included, appear scattered across sections. A dedicated subsection on data cleaning and processing would enhance reproducibility.
In the discussion, total distance emerges as the main predictor of internal load, which is consistent with previous research. However, the emphasis on novelty is overstated. The limitations section should explicitly acknowledge the absence of repeated-measures modeling and the lack of transparent thresholds for speed and acceleration.
Finally, the ethics statement requires more detail. Since the participants are minors, the manuscript should clearly indicate whether parental consent was obtained and whether approval or waiver was granted by an ethics committee. This information is essential for compliance with publication standards.
In summary, the dataset is valuable and the topic timely, but the manuscript needs substantial revision. Greater methodological transparency, more rigorous statistical treatment, and a more balanced framing of the study’s contribution are necessary before it can be considered for publication.
Sincerely,
Reviewer
Reviewer 2 Report
Comments and Suggestions for Authors
This study addresses the relationship between internal load (IL) and external load (EL) in U16 soccer players, which is relevant to optimizing training in youth athletes. The use of HR monitoring, RPE, and GPS data provides a comprehensive approach. However, the manuscript would benefit from clarification and further details in several areas to strengthen its methodology and interpretation of results.
Abstract:
- The abstract provides a good overview, but consider:
- Adding a specific key finding to highlight the main practical implication. For example, "Total distance emerged as a strong predictor of both TRIMP and HR exertion, suggesting a valuable metric for monitoring training load in this population."
Introduction:
- The introduction sets the stage well, but:
- Briefly define TRIMP, eTRIMP, RPE, and s-RPE upon their first mention for readers unfamiliar with these terms.
- Expand slightly on the limitations of traditional HR-based measures in youth athletes due to varying maturation rates. This will further justify the study's focus.
- The final sentence of the introduction is somewhat unclear. Rephrase it to more directly state the study's aims: "Therefore, this study investigates the relationship between HR-based measures, GPS-derived variables, RPE, and s-RPE in U16 soccer players to determine the feasibility of using these metrics to track training load and identify GPS variables that predict HR values."
Materials and Methods:
- Procedure:
- Explain the rationale for focusing on the competition period. Did this choice influence the type of training sessions included?
- Provide more details on the "artificial pitches" used. Were they all the same type of surface? This could impact GPS data.
- What specific steps were taken to ensure data synchronization between GPS and HR monitors?
- Why was the cool-down period excluded from the analysis?
- Briefly describe the specific technical and tactical components of the training program.
- Clarify what is meant by "data sets showing substantial deviation from these signal quality indicators were excluded from the analysis". What values are considered a substantial deviation from averages (HDOP and GNSS) and how many records were excluded after considering this aspect.
- Participants:
- State the inclusion criteria of all the players with 80% of training sessions and at least 60 minutes during the match. This way the population sample will be better understood.
- Justify the decision to exclude goalkeepers due to a lack of sensors. Were there no GPS or HR sensors available for goalkeepers, or was there another reason for their exclusion?
- The text declares that only outfield players, who fulfilled a criteria of the 80% attendance to training sessions and played 60 minutes on the match, were measured for the study. What happened to those participants who played less than 60 minutes. Is that information known to the researchers? Did the researchers took it into account?
- Explain why the team's coach was the strength and conditioning coach and not the team's physical trainer? State the number of team's coach in the participants.
- External training load measurement procedures:
- Justify the choice of a 10 Hz GPS device. Is this frequency sufficient for capturing the movements of U16 soccer players? Provide citations to support this.
- Clarify how the velocity-based variables were classified. What were the specific speed thresholds used to define MSR, HSR, and SPR? Were these thresholds based on previous research or determined specifically for this study?
- Explain, with more detail, which technology was used to measure (as a subproduct) the accelerometer.
- Why was selected the given equation to evaluate the player load? Provide the citation and discuss the scientific reasoning behind its selection.
- Internal training load measures:
- Justify the use of spiroergometric running test to evaluate the individual maximum heart rate, compared with field test. Which running test?
- How often the authors updated the new maximum heart rate after matches?
- Provide the formula for TRIMP calculation.
- Statistical analysis:
- What specific tests were used to assess normality? The Shapiro-Wilk test is mentioned, but were any other tests conducted?
- Justify the use of Spearman's rank correlation coefficient, given that the data were not normally distributed. Why not consider non-parametric alternatives?
- Clarify the criteria used for including GPS-based variables in the multiple linear regression model. What was the rationale for selecting TD, HSR, SPR, and ACC as predictors?
- Report the specific criteria used to assess collinearity (e.g., VIF values, tolerance values).
- In addition to R-squared values, report other measures of model fit for the regression analysis (e.g., adjusted R-squared, RMSE).
- What are the consequences of having removed data set after considering that there were a significant deviation to selected data.
- Clarify the unit for the parameters. In multiple places the symbol "μ" is used, instead of ±, for example for SD measure, leading to errors. The AU acronym is used to evaluate Player Load, what is its meaning. This must be clarified.
Results:
- The results section lacks clear reporting of the statistical findings. In this sense:
- Provide specific p-values for all significant correlations.
- Include confidence intervals for the correlation coefficients.
- Report standardized beta coefficients and p-values for each predictor in the multiple linear regression model.
- In Table 3, specify the units for each variable (e.g., TD in meters, HR exertion in arbitrary units).
- Figures 3 and 4 are difficult to interpret. Improve the visual presentation by:
- Increasing the font size of axis labels and variable names.
- Using a clearer color scale for the correlation coefficients.
- Adding a legend to explain the color scale.
- Were the p-values adjusted for multiple comparisons to control the risk of a type I error?
Discussion:
- The discussion does not provides sufficient detail to the finding. Therefore:
- Connect the findings back to the existing literature more explicitly. How do the results of this study compare to those of previous studies on similar age groups or populations?
- Address the limitations of the study more thoroughly.
- Discuss the potential implications of these findings for the development of training programs for U16 soccer players.
- Further expand future research directions.
Overall:
- The writing quality is good overall.
- Ensure that all references are correctly formatted and cited.
- The manuscript would benefit from a thorough proofread to correct any minor grammatical or spelling errors.
Reviewer 3 Report
Comments and Suggestions for Authors
Many thanks to the Editor for the opportunity to revise the article entitled: “Relationship Between Internal and External Load in Under-16 Soccer Players: Heart Rate, Rating of Perceived Exertion, and GPS-Derived Variables”. In this manuscript, the authors explored the relationship between external training load and internal training load among U16 soccer players.
As a main strength, the article presents a very well-structured and focused introduction, and the description of the methodology is highly accurate. Moreover, the authors have employed reliable and commercially available tools (GPS and HR), which add to the robustness of the study.
However, I have some suggestions that could be addressed to improve the manuscript, presented below:
I would suggest reducing the number of GPS-derived variables analysed, as the manuscript currently presents 8 locomotor and 8 mechanical variables. In particular, Table 3 and Figure 3 would benefit from a more selective presentation, which would enhance their readability. Among the reported variables, which ones are most commonly used to prescribe and monitor the training regimen of the soccer team under study? Once chosen, please remove them from Table 1, Table 3, and figure 3 and 4.
Could you please report the average number of field-based training sessions analysed per player? For instance, on average, how many training sessions and matches were included for each player (along with the minimum–maximum range)?
Could you insert a brief description of each variable in the Methodology section? For example, what is the cut-off point between MRS and HRS? Additionally, for ACC and DEC, does the analysis include all accelerations and decelerations, or only those greater than 3 m/s²?
I am not familiar with the metric 'HR exertion,' which is clearly reported in Table 2. Has this measure been previously employed in other studies? If so, were the results consistent with those observed in the present work?
In Table 3, can you merge the column min-max and range into one?
Regarding Figure 1, in the training sessions it appears that TD does not fully correspond to TRIMP (AU) and HR exertion during the 1st and 11th weeks, as TD increases while IL decreases (11° week) or remains quite stable (1° week). Could you explain why this occurred? In addition, would it be possible to present the weekly correlations in Figure 1? Finally, could you clarify why the correlation appears weaker during matches compared to training sessions?
Table 3. Are the results reported here for TD, HSR, MSR, and related variables comparable to those described in previous research on U16 soccer players?
Could you contextualise your results with the players’ age, age at peak height velocity, and maturity offset reported in lines 117–118? For instance, by comparing your findings with those of similar and/or older soccer players.
Do you think the correlation between ITL and ETL might vary across sessions within a microcycle? For example, I would hypothesise that ITL may correlate more strongly with ETL in high-intensity sessions (e.g., your MD-4 and MD+3 trainings) compared to technical/tactical sessions (e.g., MD-2 and MD-1). Since all sessions were merged in your analysis, could this be considered a limitation?
Finally, just a few suggestions:
Body weight could be reported with only one decimal number (e.g. 61.7± 5.2 kg instead of 61.73 ± 5.23 kg). Maybe, also, several variables in Table 3 could benefit from a reduced number after the decimal point.
Round 2
Reviewer 1 Report
Comments and Suggestions for Authors
Dear authors, thank you for submitting the manuscript for my review. After reading it, I realize that you have not changed most of the recommendations that led to the manuscript's rejection. Please note that the response only states that no changes were made, except for comments 4 and 5. Given this situation, I maintain the rejection of the manuscript until all the identified deficiencies are incorporated into the text, including modifications to the introduction, statistical procedures, and participant consent for the study.
Sincerely,
Reviewer
Reviewer 3 Report
Comments and Suggestions for Authors
The authors have clearly improved the manuscript after the revision process. I have no further recommendations.
Author Response
Thank you very much!